

# Aguhyper: a hyperledger-based electronic health record management framework

Beyhan Adanur Dedeturk and Burcu Bakir-Gungor

Department of Computer Engineering, Abdullah Gul University, Kayseri, Turkey

## ABSTRACT

The increasing importance of healthcare records, particularly given the emergence of new diseases, emphasizes the need for secure electronic storage and dissemination. With these records dispersed across diverse healthcare entities, their physical maintenance proves to be excessively time-consuming. The prevalent management of electronic healthcare records (EHRs) presents inherent security vulnerabilities, including susceptibility to attacks and potential breaches orchestrated by malicious actors. To tackle these challenges, this article introduces AguHyper, a secure storage and sharing solution for EHRs built on a permissioned blockchain framework. AguHyper utilizes Hyperledger Fabric and the InterPlanetary Distributed File System (IPFS). Hyperledger Fabric establishes the blockchain network, while IPFS manages the off-chain storage of encrypted data, with hash values securely stored within the blockchain. Focusing on security, privacy, scalability, and data integrity, AguHyper's decentralized architecture eliminates single points of failure and ensures transparency for all network participants. The study develops a prototype to address gaps identified in prior research, providing insights into blockchain technology applications in healthcare. Detailed analyses of system architecture, AguHyper's implementation configurations, and performance assessments with diverse datasets are provided. The experimental setup incorporates CouchDB and the Raft consensus mechanism, enabling a thorough comparison of system performance against existing studies in terms of throughput and latency. This contributes significantly to a comprehensive evaluation of the proposed solution and offers a unique perspective on existing literature in the field.

## INTRODUCTION

The rise of information technology has sparked a significant transformation in healthcare, shifting from article records to electronic health records (EHRs). These digitized patient documents encompass a wealth of medical information, including medical histories, demographic details, laboratory test reports, and sensitive patient data such as social security numbers (*Kruse et al., 2017*). EHRs now play a pivotal role in advancing life sciences, with ongoing exploration into innovative methods of assessing medical histories (*Al Mamun, Azam & Gritti, 2022*). For example, health data collected from smart wearable devices can monitor vital parameters, while predictive models aid healthcare professionals in evaluating patients' conditions. These advancements have significant implications for

Corresponding author
Beyhan Adanur Dedeturk,
beyhan.adanur@agu.edu.tr

public health, facilitating the anticipation and prevention of diseases before they escalate into serious threats. However, ensuring seamless continuity and operational efficiency necessitates the easy exchange of EHRs. Unfortunately, EHR sharing remains less widespread than desired.

EHRs are highly sensitive due to their containing personal information about individuals. It's understandable that individuals prioritize the protection of their privacy in this regard. In traditional systems, EHRs are stored on centralized servers (*Chenthara et al., 2019*; *Cheng et al., 2017*; *Li et al., 2016*), making the data an attractive target for intruders. Numerous studies have highlighted the increased security risks associated with centralization, requiring trust in a single authority (*Mohurle & Patil, 2017*; *Berghel, 2017*). Furthermore, service providers manage health records, leaving data owners with insufficient mechanisms for full control. Another challenge confronting the modern healthcare sector is the limited interoperability of EHRs (*Li et al., 2021*; *Adel et al., 2022*; *Aghahosseini & Sakhaei-nia, 2024*). The utilization of diverse formats and standards impedes the seamless transmission of fragmented health data among various stakeholders, hindering the integration and analysis of patient information, especially in urgent medical scenarios. The potential irreversible loss of records if an EHR is deleted from the hospital's database emphasizes the necessity for a tamper-proof system accessible only to authorized entities. Additionally, ensuring system security is essential because individuals with legitimate credentials accessing data pose significant risks to health records stored on cloud servers, surpassing those posed by external threats (*Chenthara et al., 2020*). Despite the commendable features of the existing healthcare industry, it falls short in providing a universally unified and efficient approach for storing, sharing, and analyzing health data (*Chenthara et al., 2020*; *Dedeturk, Soran & Bakir-Gungor, 2021*; *Pilares et al., 2022*).

In today's healthcare data management landscape, blockchain (BC) and the Interplanetary File System (IPFS) have emerged as powerful solutions to address challenges related to privacy, security, and interoperability (*Al-Kaabi & Abdullah, 2023*; *Divyashree & Ravi, 2023*; *Pilares et al., 2022*). Blockchain serves as an immutable and decentralized ledger, creating a chain of interconnected blocks. This distributed architecture enables participants to collaboratively make decisions without the need for a central administrator (*Tao et al., 2023*). Each block contains a cryptographic hash function of its predecessor, a timestamp, and transactional data (*Nakamoto, 2008*; *Sun et al., 2007*). Transactions undergo systematic approval by the system before being recorded onto blocks, involving active user participation in the consensus mechanism (*Dong, Abbas & Jain, 2019*). Because of its structure, BC establishes a tamper-proof infrastructure crucial for safeguarding sensitive healthcare data. Complementing BC, IPFS offers a decentralized file system enabling global computers to collaboratively store and share files within a peer-to-peer network, avoiding the drawbacks of centralized servers. Unlike traditional addressing, IPFS utilizes content-based addressing by assigning a unique hash value or Content-Identifier (CID) to each uploaded file, simplifying subsequent retrieval. This cryptographic hash, generated from the file's contents, is computed upon upload to IPFS, where files are systematically organized into objects. Integrating IPFS with BC enhances data security by storing encrypted healthcare records in IPFS and recording their

corresponding hash values in the blockchain (*Rai, 2023*). This setup strengthens resistance to tampering, improves operational efficiency, and reduces expenses associated with storing complete records on the BC.

Researchers are integrating BC and IPFS to address the pressing requirement for secure, efficient, and effective data sharing and access in the healthcare domain (*Andrew et al., 2023*; *Al-Nbhany, Zahary & Al-Shargabi, 2024*). This combined approach not only ensures the privacy and integrity of EHRs but also contributes to resolving challenges related to scalability and the lack of interoperability in existing healthcare systems. In addition to the advantages of current BC and IPFS-based EHRs sharing systems, each platform exhibits distinct weaknesses that are still awaiting solutions (*Azaria et al., 2016*; *Mcfarlane et al., 2017*; *Medicalchain, 2018*; *Al Omar et al., 2019*; *Singh et al., 2020*; *Tanwar, Parekh & Evans, 2020*; *Chen et al., 2021*; *Mantey et al., 2022*; *Sonkamble et al., 2023*; *Kaur, Rani & Kalra, 2022*). Our analysis highlights that these platforms fall short of meeting all the requirements for effectively managing EHRs. They tend to focus on specific issues rather than addressing the full scope of necessary features. To achieve comprehensive efficiency, it is vital to thoroughly examine the entire data sharing process. This examination should encompass aspects such as access control, permissions, data verification, recording, privacy-security, and user registration. Following this, the proposed system should be implemented and its performance rigorously analyzed. It is also essential to compare the proposed system with existing ones from various perspectives to ensure a thorough evaluation. While some studies assess performance within their system, others compare their systems with existing ones based on throughput and latency metrics under similar configurations. However, an innovative approach would involve comparing the proposed system with existing studies using diverse configurations beyond basic metrics. This approach allows for the identification of different factors affecting system performance and the development of new methods with different perspectives.

This research introduces AguHyper, a blockchain framework designed to enhance data exchange, health record management, and access control in the healthcare sector. AguHyper aims to address key challenges in EHR management, including access control mechanisms, interoperability, scalability, integrity, security, and privacy. By integrating Hyperledger Fabric (*Androulaki et al., 2018*) and IPFS, AguHyper seeks to surpass existing efficiency benchmarks and fill gaps in prior research. Using a permissioned BC ensures secure interactions, while IPFS tackles the challenges of centralized storage by leveraging decentralized databases. Storing hash values in the blockchain and encrypted records in IPFS achieve health record immutability, rendering the framework tamper-resistant. Our study provides a detailed examination of the system architecture and AguHyper implementation configurations, including the use of CouchDB and the Raft consensus mechanism. The experimental setup involved implementing the CouchDB database coupled with the Raft consensus mechanism (*Hyperledger-Fabric, 2023*). System performance was systematically evaluated across datasets of varying magnitudes, scrutinizing parameters such as transaction throughput, average transaction latency, and uploading-downloading time. The study concluded with a comparative analysis of the system's performance against relevant literature studies employing distinct consensus

mechanisms and database structures. Furthermore, the analysis results were supplemented with feature-based assessments to provide a comprehensive evaluation. The following summarizes the key contributions of this article:

- Examination of the entire EHR management problem is essential for achieving comprehensive efficiency. In this regard, this framework has developed a prototype that delves into BC technology, addresses gaps in prior studies, and unveils its potential applications in healthcare solutions.
- A detailed implementation and performance evaluation of the BC-based healthcare system are provided. The experimental setup included deploying the CouchDB database with the Raft consensus mechanism. The study concluded with a comparative analysis of relevant literature studies employing different consensus mechanisms and database structures. According to our investigation, no study evaluating studies from this perspective has been identified in the available literature.
- Proposed permissioned BC-based decentralized EHRs sharing architecture and smart contract design offer better performance in terms of transaction throughput, average transaction latency, and uploading-downloading time compared to existing solutions.
- Integration of a decentralized file system for off-chain data storage provides comparable performance to existing centralized database systems while offering better security against Denial-of-service (DoS) attacks, single points of failure, and improving data integrity.

The subsequent segments of this manuscript are structured as follows: "Related Work" delves into the related work, while "Background and Preliminaries" explores the preliminary components. Following this, "Architecture of the Proposed System" elucidates the architecture of the proposed framework, "Security and Functional analysis" furnishes details regarding the system's functional mechanism, and "Implementation" presents the prototype implementation of the framework. In "Performance Analysis and Discussion", the article engages in the performance analysis and discussion of the proposed framework, and finally, "Conclusions" serves as the conclusion.

## RELATED WORK

The surge in demand for online medical services has spurred the creation of various methods for sharing EHRs among healthcare entities. However, ensuring the accuracy, automation, privacy, and security of shared data is crucial for effective healthcare services. Security, automation, and scalability pose significant challenges in the healthcare sector due to the vast volume of global medical data. Traditionally, EHRs are stored on centralized servers or clouds (*Chenthara et al., 2019*; *Younis et al., 2021*), leading to vulnerabilities in applications, services, and systems. Thus, a thorough assessment of security requirements before deploying applications in the cloud and storing data is imperative. Safeguarding personal information is also paramount, as security attacks can target data during transmission, storage, and processing, risking privacy (*Namasudra, 2018*; *Tao, Cui & Iftekhar, 2024*). Various approaches have been proposed to improve

security in the sharing of EHRs, such as utilizing artificial intelligence (AI) to mitigate network security risks, as suggested by *Hooshmand & Hosahalli (2022)*. Some of these solutions focus on technical aspects, such as simulations (*Sahu et al., 2022*), while others encompass qualitative dimensions through the implementation of architectures and diverse systems. However, these methodologies encounter challenges associated with their complexity and high energy consumption, thereby constraining their effectiveness (*Deng et al., 2021*).

Blockchain technology has gained significant attention in various fields, including EHR management, due to its decentralized and immutable nature, promising secure solutions for EHRs. Since 2016, researchers have introduced several blockchain-based EHR sharing systems, aiming to address management challenges (*Junaid et al., 2022*; *Odeh, Keshta & Al-Haija, 2022*). Initially, studies between 2016–2018 focused on core development to demonstrate the feasibility of blockchain platforms in healthcare systems, covering genomic data and EHR sharing (*Dedeturk, Soran & Bakir-Gungor, 2021*; *e-Estonia, 2012*; *Azaria et al., 2016*; *Kannan & Smith, 2016*; *Mcfarlane et al., 2017*; *Medicalchain, 2018*). From 2019 onwards, studies shifted focus solely to EHR sharing, gradually reducing the emphasis on blockchain while integrating other techniques. In the studies conducted from 2019 to 2020, BC technology emphasizes the integration of cloud-based, encryption-based solutions and the evaluation of system performance (*Abul-Husn & Kenny, 2019*; *Liu et al., 2019*; *Al Omar et al., 2019*; *Niu et al., 2020*; *Tanwar, Parekh & Evans, 2020*). Between 2021 and the present, blockchain evolved into a platform for executing additional AI-based algorithms, focusing on designing blockchain-based healthcare systems with patient monitoring and disease prediction methods (*Veeramakali et al., 2021*; *Połap, Srivastava & Yu, 2021*; *Chen et al., 2021*; *Arul et al., 2021*; *Jabarulla & Lee, 2021*; *Shah & Rajagopal, 2022*; *Azbeg, Ouchetto & Jai Andaloussi, 2022*; *Jayabalan & Jeyanthi, 2022*; *Mantey et al., 2022*; *Sharma, Namasudra & Lorenz, 2023*; *Sonkamble et al., 2023*; *Yang, Li & Fan, 2023*; *Rai, 2023*; *Abdelgalil & Mejri, 2023*; *Kaur, Rani & Kalra, 2022*; *Datta & Namasudra, 2024*). This phase signifies the initial stages of building a data ecosystem using blockchain technology. In Fig. 1, the areas of focus and the problems targeted by studies from 2016 to the present have been summarized.

*Jabarulla & Lee (2021)* introduce a proof-of-concept for a distributed patient-centric image management (PCIM) system using the Ethereum blockchain and IPFS. The system aims to tackle challenges in medical image storage and sharing by offering decentralized storage and secure patient data control. It utilizes an Ethereum smart contract for distributed access control, and evaluation on an Ethereum testnet validates the efficiency and feasibility of the framework. However, issues persist regarding consumer accessibility and clarity in the data entry process. Moreover, the study overlooks considerations regarding potential data quality manipulation and lacks measures against malicious data transmission, even within an encrypted system.

*Shah & Rajagopal (2022)* proposed the M-DPS architecture for decentralized patient data management in healthcare. M-DPS aims to optimize storage, reduce gas fees, and enhance data accessibility compared to the existing DPS architecture. Evaluation results demonstrate significant improvements in gas fee reduction and storage space efficiency,

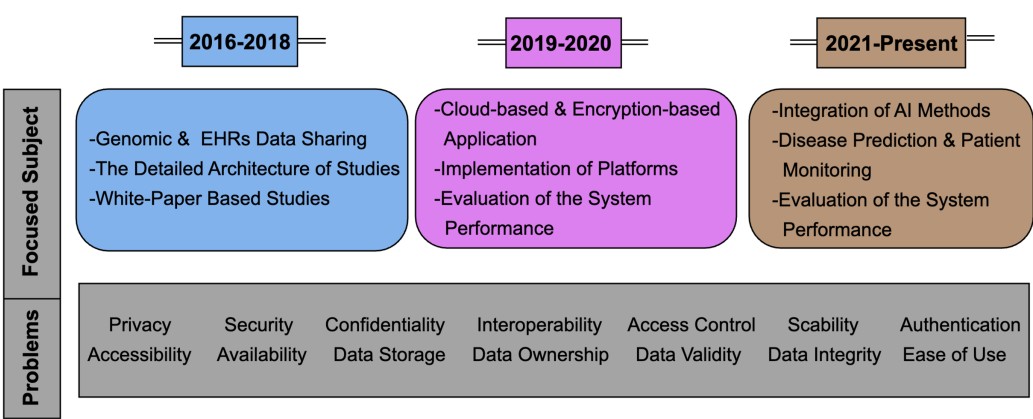

**Figure 1 The areas of focus and the problems targeted by studies from 2016 to the present.**

offering potential benefits for users. However, the study lacks detailed information on user registration processes, roles, permissions, and data exchange mechanisms, leading to a limited understanding of system operations. *Azbeg, Ouchetto & Jai Andaloussi (2022)* introduce BlockMedCare, a secure healthcare system merging IoT with blockchain to address IoT-driven healthcare needs. Focused on remote patient monitoring, the system employs a re-encryption proxy and blockchain for security, smart contracts for access control, and an off-chain IPFS database for scalability. A diabetes management use case demonstrates the system's effectiveness, with experimental results presented through system interfaces. However, the study lacks details on the data input process and fails to consider scenarios where malicious entities could transmit irrelevant data, resulting in a lack of mechanisms for data verification.

*Kaur, Rani & Kalra (2022)* address the complexities of managing EHRs distributed among multiple healthcare providers by proposing a permissioned blockchain-based framework. This framework utilizes Hyperledger Fabric for network implementation, IPFS for secure off-chain storage of encrypted data, and the Identity Based Proxy Re-Encryption (IB-PRE) algorithm for secure data sharing. The framework's efficiency is tested using performance testing with Hyperledger Caliper. Comparative analyses with existing solutions show its effectiveness in addressing EHR security and privacy concerns. Although the study offers a relatively comprehensive performance analysis compared to its counterparts, it lacks diversity in workload perspectives. The experimental setup mirrors that of comparative studies, focusing solely on evaluations related to latency and throughput.

*Sonkamble et al. (2023)* present a patient-centric healthcare data management system based on Hyperledger Fabric. This decentralized architecture emphasizes patient control, ensuring secure storage of EHR data through IPFS and BC integration. Secure password authentication-based key exchange (SPAKE) facilitates user control through smart contracts. The experimental setup demonstrates the system's effectiveness in patient-centric access control and conducts performance assessment based on key parameters. The study evaluates the access control mechanism while benchmarking performance against

existing studies. However, a more comprehensive analysis, particularly regarding blockchain-based performance, could have provided a nuanced comparison with existing solutions. Additionally, it remains unclear how consumers would interact with or engage in these systems built on the Hyperledger framework.

*Sharma, Namasudra & Lorenz (2023)* utilize smart contracts to enhance security features for cloud-stored data. Their proposed method encrypts data before uploading it to the cloud server, ensuring confidentiality. Additionally, the system incorporates an improved optimization technique and a challenge-response-based integrity verification mechanism to bolster cloud environment security. Furthermore, it aims to reduce system bandwidth costs through an enhanced fruit fly optimization algorithm, streamlining node failure repair processes. Despite the utilization of blockchain, the study lacks blockchain-specific analysis in the performance assessments, and it does not provide detailed information about blockchain aspects of system implementation. Moreover, specifics regarding data sharing and system permissions are not available.

In recent years, the integration of BC with mobile edge computing (MEC) has bolstered efficiency in healthcare by providing high computing power in close proximity to users. While various blockchain-based MEC approaches have been proposed to address EHR security issues, many still face challenges in efficient implementation and fail to tackle scalability and automation concerns (*Wang et al., 2022*). *Datta & Namasudra (2024)* introduce a novel blockchain-based EMR sharing framework that utilizes MEC and consumer electronic devices to enhance existing schemes. This framework incorporates additional security layers through techniques such as AES. Encrypted EMRs and diagnosis reports are stored in IPFS storage, with corresponding hashes uploaded to the blockchain network. Smart contracts manage different functionalities, and the proof of authority (PoA) consensus algorithm ensures faster transactions. However, the study lacks details on the data input process and does not address scenarios where malicious entities may transmit irrelevant data, resulting in a lack of mechanisms for data verification. Additionally, detailed information on system permissions is not provided.

The studies on EHR sharing conducted from 2016 to the present are systematically summarized and compared, utilizing key metrics presented in Table 1. Each platform exhibits distinct strengths and weaknesses. Our analysis reveals that these platforms have effectively addressed significant challenges in EHR sharing. However, they often concentrate on particular issues rather than encompassing the complete array of required functionalities. To attain comprehensive effectiveness, it is essential to meticulously scrutinize the entire data sharing process, such as access control (data interoperability), permissions, data verification, data recording, data input (privacy and security), and user registration (roles). Subsequently, the proposed system should be implemented, followed by an exhaustive performance analysis. Furthermore, it is imperative to compare the proposed system with existing systems from various perspectives to ensure a comprehensive evaluation. In our findings, although the studies are blockchain-based, some of the studies lack blockchain-specific analysis in performance evaluation. Some other studies conduct performance analysis within their own systems, while others compare their systems with existing ones based on throughput and latency metrics.

**Table 1 Comparison of features between the proposed work and existing related works.**

| Research Work | ACM | P | DV | SP | Roles | DS | S | A | PA | DP |
|---|---|---|---|---|---|---|---|---|---|---|
| *e-Estonia (2012)* | √ | √ | X | √ | √ | √ | N/A | √ | X | X |
| *Azaria et al. (2016)* | √ | √ | X | √ | √ | √ | X | X | X | X |
| *Kannan & Smith (2016)* | √ | X | X | X | X | √ | X | X | X | N/A |
| *Mcfarlane et al. (2017)* | √ | X | X | X | X | √ | √ | X | X | X |
| *Medicalchain (2018)* | √ | √ | X | √ | √ | √ | X | X | X | X |
| *Abul-Husn & Kenny (2019)* | X | √ | X | X | X | √ | X | X | X | X |
| *Liu et al. (2019)* | X | √ | X | X | X | √ | X | X | √ | X |
| *IBM Medical Blockchain (2019)* | √ | √ | X | X | √ | √ | √ | X | X | X |
| *Al Omar et al. (2019)* | √ | √ | X | X | √ | √ | √ | X | √ | X |
| *Niu et al. (2020)* | √ | X | X | X | √ | √ | √ | X | X | X |
| *Tanwar, Parekh & Evans (2020)* | √ | √ | X | X | √ | √ | √ | √ | √ | X |
| *Veeramakali et al. (2021)* | N/A | X | X | X | X | √ | X | X | X | √ |
| *Połap, Srivastava & Yu (2021)* | √ | √ | X | √ | √ | X | X | X | X | √ |
| *Chen et al. (2021)* | √ | √ | X | X | √ | X | √ | √ | X | √ |
| *Arul et al. (2021)* | √ | √ | X | X | X | √ | √ | X | X | √ |
| *Jabarulla & Lee (2021)* | √ | √ | X | X | √ | √ | √ | √ | √ | X |
| *Shah & Rajagopal (2022)* | √ | X | √ | √ | X | X | √ | √ | √ | X |
| *Azbeg, Ouchetto & Jai Andaloussi (2022)* | √ | √ | X | X | √ | √ | √ | √ | √ | X |
| *Jayabalan & Jeyanthi (2022)* | √ | √ | X | X | √ | √ | √ | √ | √ | X |
| *Mantey et al. (2022)* | X | X | X | X | X | X | √ | X | X | √ |
| *Sharma, Namasudra & Lorenz (2023)* | √ | X | √ | √ | √ | X | √ | √ | X | X |
| *Sonkamble et al. (2023)* | √ | √ | X | √ | √ | X | √ | X | √ | X |
| *Yang, Li & Fan (2023)* | √ | X | X | X | √ | √ | X | X | X | X |
| *Rai (2023)* | √ | √ | X | √ | √ | √ | √ | √ | X | X |
| *Abdelgalil & Mejri (2023)* | √ | √ | X | √ | √ | √ | √ | √ | X | X |
| *Kaur, Rani & Kalra (2022)* | √ | √ | X | √ | √ | X | √ | √ | √ | X |
| *Datta & Namasudra (2024)* | √ | X | X | √ | √ | √ | √ | √ | √ | X |
| Proposed model | √ | √ | √ | √ | √ | √ | √ | √ | √ | √ |

-ACM: Access control mechanism    -P: Permissions    -DV: Data verification

-SP: Security and privacy    -DS: Data sharing    -S: Scability

-A: Availability    -PA: Performance analysis based on BC    -DP: Disease prediction

However, an innovative approach would involve comparing the proposed system with existing studies based on diverse configurations beyond basic metrics. This comparison method enables the identification of various factors influencing system performance and the creation of novel approaches from diverse viewpoints.

This study establishes a patient-centric interoperability framework, leveraging a hyperledger fabric-based blockchain network using Hyperledger Composer and IPFS for secure and controlled storage of EHRs. The outlined framework guarantees patients

comprehensive control, encompassing aspects such as security, privacy, scalability, and data integrity. To enhance data efficiency, the approach involves storing solely the hash of health records on the BC, while the bulk of the data is encrypted and stored off-chain in the IPFS. The study includes a prototype that examines BC technology, addresses prior gaps, and highlights potential healthcare applications. Comprehensive coverage of system architecture and AguHyper implementation, along with performance evaluations using various datasets, is presented. The experimental setup involves CouchDB and Raft consensus. The study concluded with a comparison of the system's performance against previous research in the field, where diverse consensus mechanisms and database structures were employed. Furthermore, the analysis results were enhanced by incorporating feature-based assessments, contributing to a comprehensive and detailed evaluation. Remarkably, our research indicates that no study has undertaken a holistic examination of the processes in blockchain-based EHR sharing platforms, comparing the performance of the AguHyper with existing studies across different configurations and perspectives. We believe that this study will be done by drawing the attention of scientists in various fields to this area and by enabling them to develop new approaches to solve the problems raised by these issues.

## BACKGROUND AND PRELIMINARIES

The subsequent section provides a succinct elucidation of the foundational components inherent in our proposed framework.

### Hyperledger fabric

Selecting the most appropriate blockchain platform for the conceptualization and development of a blockchain-centric project constitutes a crucial undertaking. Two main categories of BCs, namely public and private, exist. Public blockchains are designed for complete transparency and permissionless access, allowing anyone in the network to access the transaction ledger and perform operations without restrictions. Conversely, private blockchain technology is tailored to fulfill the requirements of applications prioritizing privacy and security (*Androulaki et al., 2018*). By adjusting access permissions on the network, a closed network can be easily established, and multiple channels can be created, restricting usage to specified users. This ensures that unregistered users cannot access the ledger, and private information can be shared within the network without notification (*Iftekhar et al., 2021*). In light of the sensitive nature of EHRs and the imperative for controlled access, our study employs Hyperledger Fabric. This choice ensures the secure sharing of healthcare information among pre-defined parties, eliminating the need for dependence on a central authority.

### Consensus mechanism

A foundational aspect and stratum of blockchain is the consensus mechanism governing transactions. This mechanism relies on the smart contracts layer to authenticate and modify transactions within the ledger in the sequence of their occurrence. Within the ledger, the consensus protocol dictates the transaction order and the rejection of

suboptimal transactions. Hyperledger Fabric encompasses three distinct implementations of the consensus algorithm (*Hyperledger-Fabric, 2023*; *Zheng et al., 2017*): i) SOLO ordering service: This is a deployable nonproduction ordering service, featuring a single central authority and a solitary process catering to all clients, obviating the need for consensus. While suitable for development and testing, it is not recommended for deployment. ii) Kafka-based ordering service: Built on Kafka's publish-subscribe architecture with multiple Kafka brokers and respective Zookeeper ensembles, this service provides crash-fault tolerance (CFT). Despite storing data on other brokers in the event of a failure, it lacks Byzantine fault tolerance, offering no defense against malicious nodes on the network. iii) Raft: As a CFT ordering service, Raft is based on the Raft protocol in etcd. In the Raft protocol, which operates on a "leader and follower" model, a leader node is elected for each channel, and the followers replicate its decisions. Raft ordering services are anticipated to be more straightforward to set up and manage than Kafka-based ordering services, allowing diverse organizations to contribute nodes to a distributed ordering service. Given the attributes of these three consensus mechanisms, the Raft mechanism has been chosen for our study. This decision aligns with our system requirements, and our intention is to conduct a performance analysis and comparative evaluation of Raft with other consensus mechanisms employed in existing systems.

### State database

Hyperledger Fabric offers support for two peer database formats: CouchDB and LevelDB. LevelDB, functioning as a key-value store, stores chaincode data in a simple format, enabling the execution of key, key range, and composite key queries. On the other hand, CouchDB utilizes a JSON-formatted datastore, providing greater flexibility by allowing the mapping of information between different database documents (*Ndzimakhwe et al., 2023*). For this study, CouchDB is specifically chosen as the on-chain database. Its utilization contributes not only to the security and data protection aspects of the system but also enhances system compliance. The JSON format of CouchDB allows for a more dynamic and versatile representation of data within the blockchain, aligning with the requirements and objectives of the research.

### Hyperledger composer

Hyperledger Composer is a suite of collaborative tools devised for the design and modeling of blockchain business networks. Its purpose is to streamline and expedite the process for business owners and developers in the creation of smart contracts and blockchain applications. Hyperledger Composer was designed with the objective of simplifying the challenges associated with direct engagement with Hyperledger Fabric. It offers a more elevated interface, enabling developers to articulate their business networks, participants, assets, and transactions with greater ease. This encompasses the provision of a modeling language, an API, and a suite of command-line tools to facilitate and enhance the development workflow (*Mali et al., 2023*; *Dhillon, Metcalf & Hooper, 2017*). For this reason, Hyperledger Composer is used in our study. In the context of this research, Composer is employed to generate a business network definition. This definition includes

model files (.cto) that specify assets, script files (.js) containing smart contracts, ACL (.acl) files for access control rules and permissions, and Query (.qry) files for formulating database queries within the framework. Subsequently, the business network definition is encapsulated into a .bna file to facilitate the deployment of the framework's business network onto a distributed ledger.

### Chaincode

Smart contracts, serving as autonomous chain codes, encapsulate the regulations dictating particular network transactions. In Hyperledger Composer, these smart contracts are scripted in JavaScript and carried out on the Hyperledger Fabric blockchain network. The chaincode implemented in Hyperledger Composer serves to embody the application logic responsible for specifying and managing transactions, overseeing asset management, and enforcing access control policies within a business network (*Sasikumar & Karthikeyan, 2023*). In the AguHyper project, the decision to utilize smart contracts is deliberate, leveraging their inherent benefits, including the automated execution of contractual obligations and the effective regulation of data access permissions and relationships.

### Interplanetary file system

IPFS is a decentralized, peer-to-peer file system designed to revolutionize the current web structure, potentially replacing HTTP. When utilizing IPFS to access a data structure or retrieve a file from the web, the process involves retrieving it through network peers using the file's unique identifier, or 'cryptographic hash'—a feature known as IPFS content addressing (*Benet, 2014*). If the data surpasses a predefined size threshold, IPFS ensures secure storage by distributing the encrypted data across multiple nodes. In the context of this study, IPFS serves as an off-chain database for storing an extensive array of healthcare records and their corresponding hash stored in the CouchDB database (*Liu & Wang, 2023*).

## ARCHITECTURE OF THE PROPOSED SYSTEM

Within this section, we present the envisioned architectural framework based on Blockchain, as depicted in Fig. 2. This framework delineates three discrete layers: the Storage layer, the Blockchain layer, and the User layer. The User layer encompasses potential participants in the BC network. Before people become system users, they enter the necessary information into the system using Client APP. This registration requests are transmitted to the Blockchain layer *via* API. The Blockchain layer produces a digital signature by assigning a public-private key to the person by the certificate authority (CA). Once the user has a digital signature, the MSP (system organization is called AguHyper) stores the user's digital identity and permissions according to their role in the system. After these stages, the person becomes a system user according to the relevant role. Users can input data, request data and share data to the Blockchain layer through the client APP according to their roles and permissions. The Storage layer integrates an off-chain distributed file system explicitly engineered to house users' encrypted data. This data is systematically organized and referenced through corresponding hashes. A user with data

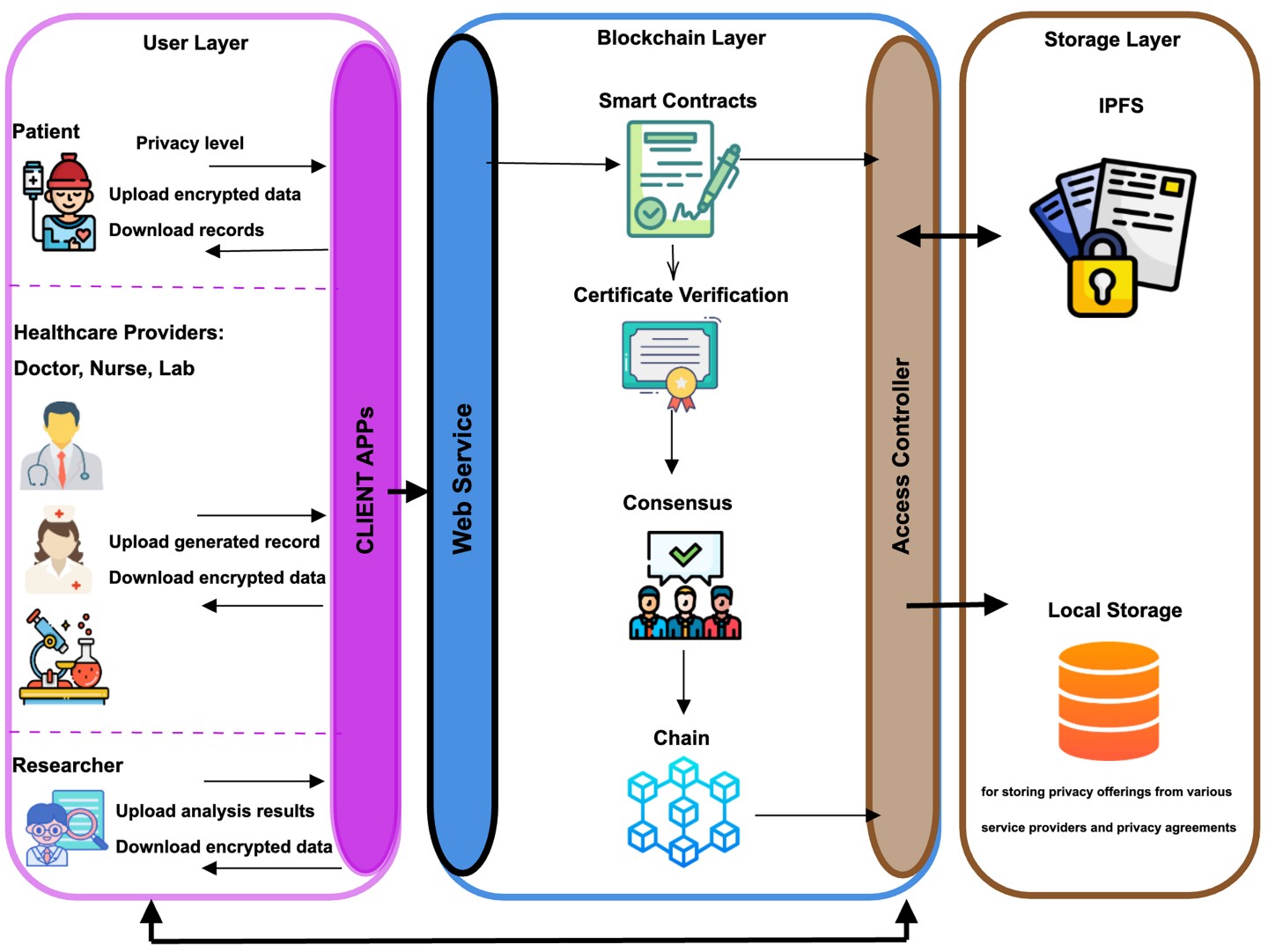

**Figure 2 Architecture of AguHyper.** Figure source credits: https://www.flaticon.com/free-icon/patient_706161; https://www.flaticon.com/free-icon/doctor_3774299; https://www.flaticon.com/free-icon/nurse_119044; https://www.flaticon.com/free-icon/researcher_2247876; https://www.flaticon.com/free-icon/laboratory_8711362; https://www.flaticon.com/free-icon/blockchain_2091665; https://www.flaticon.com/free-icon/consensus_7179065; https://www.flaticon.com/free-icon/certificate_3885250; https://www.flaticon.com/free-icon/contract_1358615; https://www.flaticon.com/free-icon/database_657695; https://www.flaticon.com/free-icon/key-lock_6169756.

entry authority makes a request to the Blockchain layer. The relevant encrypted data is recorded in IPFS by the authorized organization in the Blokchain layer using the APIs of the IPFS layer with user information. After this process, the hash of the relevant data is recorded on the BC. The Blockchain layer is tasked with preserving metadata and ownership details pertaining to files stored in the decentralized file storage. Moreover, it provides permission management services, fostering secure data sharing among entities. While communication between each layer is provided through APIs, the Blockchain layer is the basic layer in communication between the User layer and the Storage layer.

## Storage layer

Instead of utilizing BC for the storage of healthcare data, we have chosen to employ the IPFS to store encrypted data blocks. It is noteworthy that IPFS operates without a singular point of failure and possesses the capability to efficiently disseminate substantial amounts of information without redundancy (*Shuaib et al., 2022*). Users generate EHRs and store them in a distributed manner across IPFS storage nodes. Each file uploaded to the IPFS system is assigned a unique hash string, facilitating its subsequent retrieval. When integrated with the BC network, the IPFS system ensures data integrity. After storing the data, the storage node transmits the hash of the data to the BC network. This mechanism enables the straightforward detection of any unauthorized modifications.

## User layer

Every user deploys a decentralized application specifically designed to facilitate interaction with both the BC and the distributed file system. Within the system, a CA generates public-private key pairs for users and creates a digital signature for each user using their private key, which also includes the user's public key. Once a user obtains a digital signature, the MSP stores the user's digital identity and permissions based on their role in the system. Simultaneously, the MSP system maintains a folder containing the list of digital signatures owned by users. When a transaction occurs, it is signed with the private key of the client initiating the transaction. Orderer nodes play a role in processing this transaction onto the blockchain. The transaction undergoes verification with the client's public key according to the relevant consensus mechanism before being processed onto the blockchain.

In our system, only patients, labs, nurses, and doctors are authorized to input data into the system. Meanwhile, researchers and doctors have the privilege to submit data requests.

- **Patients:** Each patient node assumes responsibility for the management of one or more EHRs. They transmit their encrypted data to the IPFS storage node. These nodes exhibit the capability to generate and disseminate transactions. EHR access permissions are entirely under the control of the patients.
- **Hospitals:** Nodes serve as system users responsible for registering new members, collecting transactions shared on the platform, and recording them on the blockchain.
- **Doctors:** have the capability to request data from the system. They also exhibit the capacity to securely convey encrypted EHRs to the designated storage node.
- **Researchers:** are individuals who submit requests for data and subsequently share the results of their analyses on that data.
- **Nurses and laboratories:** are users who exhibit the capacity to securely convey encrypted EHRs to the designated storage node.

## Blockchain layer

The conceived system is rooted in a permissioned BC framework, wherein a pre-defined group of nodes operates as miners. These nodes, recognized as trustworthy by the broader network, are tasked with the validation of transactions and the generation of new blocks. In

our specific context, the entities bestowed with this responsibility are reputable hospitals. These trusted authorities undertake several functions, encompassing the addition of data to the decentralized file system, uploading associated transactions to the Blockchain, and validating various transactions initiated by external users, such as requests for permission and permissions granted.

### Smart contracts

The Blockchain layer comprises three types of smart contracts: participantCreation contract, assetCreation contract, and dataSharing contract.

**participantCreation contract**: To safeguard the system against malicious users attempting to introduce inaccurate data or exploit information, all users are registered anonymously within the participantCreation contract. This registration includes user public keys and their corresponding roles. Algorithm 1 presents the pseudocode, elucidating the steps of the participant creation process.

**assetCreation contract:** The assetCreation contract maintains a record list that outlines the association between users and their respective data. Each entry in this list includes the public key of the data owner and the hash of the encrypted data, referencing the raw data stored off-chain. To streamline this process, the data contract offers functional interfaces for the addition of data. Algorithm 2 presents the pseudocode, elucidating the steps of the asset creation process.

**dataSharing contract:** A dataSharing contract meticulously documents access permissions, defining the diverse privileges held by users with respect to data housed within the dataSharing contract. Each access permission is composed of three tuples: the public key of the permission granter, the public key of the permission requester, and the hash and ID of the data. Algorithm 3 presents the pseudocode, elucidating the steps of the data sharing process.

## System operation details

### Add records

Users conduct four primary operations to add records to the system. These operations include i) uploading data to the IPFS and ii) sending metadata to the BC. During the data uploading process to IPFS, the data undergoes encryption, and the hash value is derived from the encrypted data. The upload procedure is finalized by saving the encrypted data. In the metadata sending process to BC, the transaction content is initially generated. This content encompasses pertinent information, including the encrypted key. Subsequently, the transaction is authenticated through the user's key and transmitted. In the supplementary EHRs add-on process, the data entry procedures for healthcare providers, who exclusively input patient data, differ from those performed by the patients themselves. Notably, there is an absence of an encrypted key in the content of the patient transaction.

### Data sharing request

Upon the availability of metadata on the BC, medical practitioners or researchers with an interest in specific data can initiate a permission request within the Blockchain network.

---

**Algorithm 1** participantCreation contract.

**Input:** userPublicKey, userRole

**Output:** success of Registration

1:   // The MSP register the user in the system with the necessary permissions and roles after approving the digital signature by the CA.

2:   **if** protectSystemFromMaliciousUsers() == True **then**

3:   anonymouslyStoreUserDetails(userPublicKey, userRole);

4:   **return** "SUCCESS";

5:   **else**

6:   **return** "USER CREATION ERROR";

7:   **end**

---

**Algorithm 2** assetCreation contract.

**Input:** userPublicKey, encryptedDataHash

**Output:** success of Data Addition

1:   //Allow data entry if the digital signature of the person who wants to upload data is matched with the digital signature registered in MSP.

2:   **if** SystemUsersVerify() == True **then**

3:   record = createRecord(userPublicKey, encryptedDataHash);

4:   addRecordToUserList(record);

5:   **return** "SUCCESS";

6:   **else**

7:   **return** "DATA ADDITION ERROR";

8:   **end**

---

**Algorithm 3** Sharing contract.

**Input:** userPublicKey, requesterPublicKey, EncryptedDataHash, dataID

**Output:** success of Permission Granting

1:   //Share the relevant information with the requester, İf the integrity of the data is verified and the data owner accepts the request

2:   **if** DataIntegrityVerify() == True && PermissionAcceptedbyUser()== True **then**

3:   permission = createPermission(userPublicKey, requesterPublicKey, EncryptedDataHash, dataID);

4:   addPermissionToDataSharing(permission);

5:   **return** "SUCCESS";

6:   **else**

7:   **return** "PERMISSION GRANTING ERROR";

8:   **end**

---

This is accomplished by submitting a transaction that triggers the activation of the dataSharing contract.

Upon the transmission of a permission request to the dataSharing contract for accessing specific data, the data owner receives a notification and is afforded the option to either grant or deny the request. In the event of authorization, a transaction is generated, encapsulating the subsequent components: the ID of the requested data, the public key of the requester, and the key designated for decrypting the requested data, encrypted with the public key of the requester. Post permission approval, the user retrieves the data from a nearby IPFS node. Subsequently, the retrieved data undergoes decryption.

### Analysis result share

While designing AguHyper, data sharing was considered for two different users. The first is sharing with the doctor, and the second is data sharing with researchers. Researchers are users who need data for analysis, such as disease prediction. If a data request from these users is approved, these users share the results of their analysis, such as disease prediction, with the relevant patient.

## SECURITY AND FUNCTIONAL ANALYSIS

In this section, the role of the patient is employed to elucidate the functional mechanism of the system, as depicted in Fig. 3. Initially, a patient initiates a registration request within the system. Subsequent to a meticulous evaluation of the request and satisfaction of necessary conditions, the authorized hospital grants approval. Consequently, the patient is furnished with a digital signature certificate for utilization within the system. Subsequently, the patient endeavors to input data into the system. The system has stipulated specific formats for individual data entries, necessitating the patient to adhere to format guidelines pertinent to the data type during the entry process. In this way, incorrect data entry can be prevented. EHRs are highly sensitive due to their containing personal information about individuals. It is understandable that individuals prioritize the protection of their privacy in this regard (*Wang et al., 2020*). If shared data remains in its original form on the platform, direct user access is facilitated; however, this compromises data privacy. Hence, it is imperative to maintain data privacy within the system. To ensure data privacy, EHRs are saved encrypted to IPFS instead of being saved originally. If encrypted data were stored directly in the blockchain instead of IPFS, it would create a scalability problem due to the size of the data. In order to solve both scalability and availability problems, it was preferred to keep encrypted data in IPFS and hash values securely stored within the BC because IPFS stores content on a distributed network of nodes. This architecture enhances availability because content can be retrieved from multiple nodes, even if some nodes are offline or experiencing issues. Both the decentralized structure of BC and IPFS also mitigate the risk of single points of failure because there is no central authority or server that, if compromised, could disrupt the entire system. In the system, both recipients require detailed information about the data, and the data must be classified after the data entry stage. To meet these requirements and facilitate easier browsing of data on the platform by recipients, patients are required to enter basic and general information about the data into

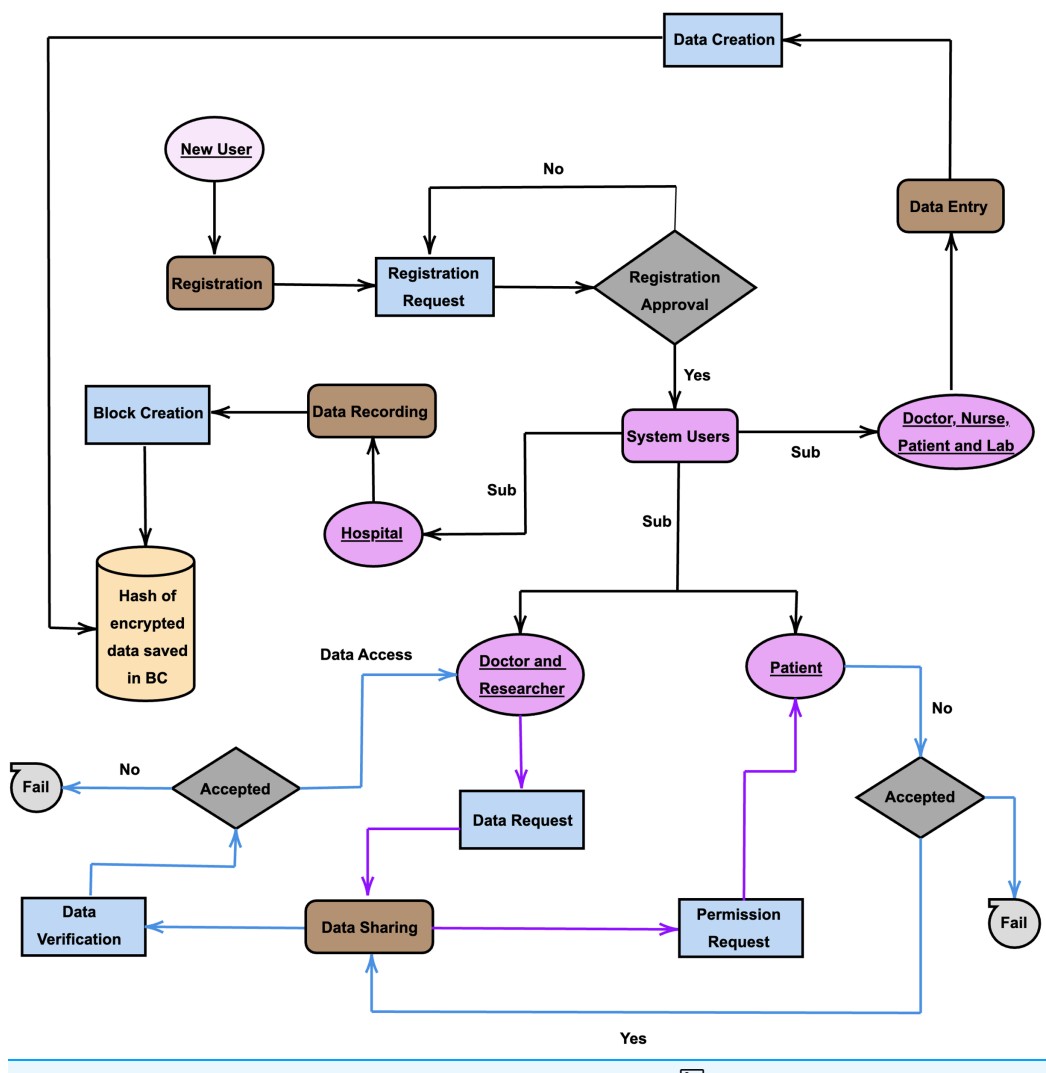

**Figure 3** Activity diagram of AguHyper.            

the data header during the data entry process. Following the entry of data header information in accordance with specified criteria, the system categorizes the data.

Upon the completion of data entry procedures, the imperative arises to systematically store the data within the system, thereby instigating the formation of blocks. The data recording process involves a meticulous scrutiny of data integrity and validity, culminating in the transformation of data into blocks. Concurrently, the data recording phase assumes a pivotal role in identifying the entities responsible for processing the data within the blocks and determining the specific consensus mechanism to be employed throughout the data recording process. The integrity of healthcare data is upheld through the exploitation of the immutability characteristic inherent in BC technology (*Conti et al., 2018*) and IPFS. In our system, the verification process entails a meticulous comparison between the hash of the encrypted data stored on the ledger and the hash applied to the encrypted data retrieved from storage. Consistency in these hashes expedites the furnishing of data to the requester, affirming its integrity. Conversely, a discrepancy in the hashes signifies potential

data corruption, triggering notifications to users. During the data recording phase, the patient's data undergoes segmentation into blocks, after which the patient proceeds to the subsequent data sharing phase.

Receivers can view the records of the data created on the platform *via* the BC. When a doctor or researcher wants to examine one of these data, they contact the relevant patient through the dataSharing contract. In the event that a doctor or researcher wishes to acquire specific data from a patient, the corresponding request is communicated to the patient. Upon receiving this request, if the patient consents to providing access to the encrypted data stored in IPFS, the requester is duly authorized to access both the hash and key associated with the data, in conjuction with their own identity. All transactions executed on the platform are periodically recorded in blocks and integrated into the chain following approval through the consensus protocol.

Authorized users (*Xu et al., 2019*) actively conduct external audits to validate the authenticity of health records. Within our system, both patients and healthcare providers share accountability for their data, as transactions include the user's signature. This ensures the unquestionable origin of data generated by a user. Implementing data access control and auditing data usage through smart contracts helps resolve medical disputes by accurately identifying responsible parties in cases of potential violations, thereby ensuring accountability. In addition to these activities, BC employs several mechanisms to mitigate the risk of DoS attacks, which aim to disrupt the availability of a network or service. Firstly, BC's decentralized nature distributes control and data across a network of nodes, eliminating single points of failure and reducing the effectiveness of traditional DoS attacks targeted at centralized systems. Secondly, consensus mechanisms require nodes to validate and agree on the validity of transactions before they are added to the BC. This agreement process prevents malicious actors from overwhelming the network with fraudulent transactions, as a majority of nodes must reach consensus for a transaction to be considered valid. Finally, the system registration process is carried out with authorized hospital grants. Operations that can be performed in the system are limited according to user roles.

## IMPLEMENTATION

For the practical execution, the Hyperledger Composer Business Network (*Dhillon, Metcalf & Hooper, 2017*) and IPFS were established and subjected to testing, and the network's performance was demonstrated under various workloads. In this section, details of the framework implementation will be given. The Hyperledger Composer serves as a development framework designed to streamline the process of creating applications for the Hyperledger Fabric blockchain. Its primary objective is to assist users in developing blockchain applications on Hyperledger Fabric without necessitating an in-depth understanding of the intricate details associated with BC networks. In addition to this, it includes a web-based platform known as the Hyperledger Composer Playground (*Hyperledger Foundation, 2023a* (Playground tutorial)), facilitating the configuration, deployment, and testing of a business network directly within a browser, eliminating the need for a local network setup.

Composer utilizes its proprietary object-modeling language to define four types of resources: i) Assets: Represent items under observation within the application, ii) Participants: Denote entities engaged in interactions within the network, each possessing its own set of permissions, iii) Transactions: Dispatched to update either an asset or a participant, as well as to execute custom-defined logic, and iv) Events: Emanate from transaction logic and can be subscribed to by participants. To harness the aforementioned advantages, this study established a Hyperledger Composer Business Network named AguHyper. AguHyper's configuration, deployment, and testing were conducted using the Hyperledger Composer Playground. AguHyper Business Network, comprised of three distinct files: model, script, and access control. The model file encompasses definitions for assets, participants, transactions, and events. The script file contains transaction logic in the form of functions, while the access control file delineates the permissions assigned to assets, participants, and transactions.

In the AguHyper: i) Participants: are patient, doctor, researcher, nurse, and lab. Hospitals are the system administrator itself, ii) Assets: are PatientData, and iii) Transactions: are ParticipantCreation, assetCreation, DataSharingDoctor and DataSharingResearcher. The "ParticipantCreation" transaction involves creating a participant by gathering the necessary information for system registration from the users. The "assetCreation" transaction encompasses the creation of an asset, wherein the encrypted data hash and requisite information for asset creation are collected from the users. During the design of AguHyper, data sharing was considered for two distinct user scenarios. The first involves sharing data with doctors, and the second involves sharing data with researchers. In the "DataSharingResearcher" transaction, researchers express the need for and request data for analysis purposes, such as disease prediction. Upon approval of the data request, relevant information about the data is shared with the researchers. Following the data sharing, researchers then share analysis results, such as disease predictions, with the corresponding patient. In the "DataSharingDoctor" transaction, doctors engage in diagnosing diseases, among other tasks. In a similar fashion to data sharing with researchers, doctors request data, and upon approval, pertinent information about the data is shared with the doctors. Permissions on assets, participants, and transaction in the system are as follows:

- Patients can read doctor and researcher information.
- Patients have full access to their assets.
- Patients can read data request transactions.
- Researchers and doctors can read the meta data of assets.
- Nobody can access the hash of encrypted data except its owners.
- Researchers and doctors can submit data request transactions.
- If the appropriate conditions are provided, the researcher and doctors get the permission to read the hash of encrypted data and the necessary information about the relevant data.

To assess the system's performance, we employed the Hyperledger Composer REST server (*Hyperledger Foundation, 2023c* (Hyperledger Composer)) through various API

calls. The Hyperledger Composer Rest Server facilitates the creation of a REST API from a deployed Hyperledger Fabric business network, offering ease of consumption by HTTP or REST clients. The Hyperledger Composer REST server executes Create, Read, Update, and Delete (CRUD) operations, enabling the manipulation of asset and participant states and facilitating the submission or retrieval of transactions through queries. For API calls, custom Node.js codes were utilized.

# PERFORMANCE ANALYSIS AND DISCUSSION

This section evaluates the effectiveness of the suggested architecture through multiple API calls on the Hyperledger Composer REST Server (*Hyperledger Foundation, 2023c* (Hyperledger Composer)), as determined by a range of experiments. To gauge the efficiency of the proposed framework, a scenario involving data sharing between healthcare professionals and patients was enacted. The key metrics employed for performance assessment include transaction throughput measured in transactions per second (tps), average transaction latency in seconds, and the time taken for data uploading and downloading (*Hyperledger Foundation, 2023b* (Hyperledger: Blockchain Performance Metrics)). The System Under Test (SUT) blockchain finalizes legitimate transactions at a specific frequency within a defined timeframe, known as transaction throughput. It is important to note that this metric encompasses the aggregate performance across all nodes within the SUT rather than focusing solely on individual node activity. On the other hand, transaction latency provides a holistic assessment of the duration required for a transaction's impact to become functional throughout the network. This evaluation encompasses the time interval from when the transaction is initially submitted to when its outcome achieves widespread accessibility across the network. Such assessment incorporates factors like propagation duration and any settlement periods influenced by the prevailing consensus mechanism.

## Experimental setup

The study introduced a Hyperledger Composer Business Network named "aguhyper". Configuration, deployment, and testing of aguhyper were conducted using the Hyperledger Composer Playground. Our implementation involved the development of custom Node.js code to invoke two chaincodes: EHRs-Data-Creation and Data-Sharing. To facilitate a comprehensive evaluation, aguhyper was configured twice, allowing for a comparative analysis with SOLO-based studies in the existing literature. In the initial setup, aguhyper utilized three peer nodes within a single organization, whereas the second setup involved one peer node for each of the two organizations, resulting in a total of two organizations. The entire system operated on an Intel Core-i9-9900K-16 CPU, 32 GB of memory, and a 500 GB storage-enabled server. Ubuntu 18.04 was chosen as the operating system for its compatibility with Hyperledger Fabric 1.4. The fabric block size was configured to 256 MB, and CouchDB was selected by Hyperledger Fabric as the world-state database. Additionally, IPFS v0.4.22 was employed for IPFS-based experiments. To enact various use cases for performance benchmarking, the study adopts distinct network configurations.

**Table 2 System configuration and simulation parameters for phase 1.**

| Phase 1: | Configuration |
|---|---|
| Processor | Intel core-i9-9900K-16 CPU |
| Memory | 32 GB |
| OS | Ubuntu 18.04 |
| Hyperledger fabric | v1.4 |
| Rounds | 10 |
| Transactions | 100, 250 and 500 |
| Transaction send rate (tps) | 5, 25, 50, 75, 100 |
| State DB | CouchDB |
| Orderer and size | Raft and 2 Org-1peer each |

### Scenario 1

The initial phase aims to comprehend the influence of altering the number of transactions (Tx) and rate (TPS) on both throughput and average latency. The network settings for the first phase are detailed in Table 2. During the measurement period, adjustments were made to the transaction rates for each respective transaction group. Figure 4 illustrates an enhancement in system throughput as the transaction per second (tps) rate increases. Nevertheless, system throughput experiences a decline as the number of transactions increases while maintaining the current TPS rates. Figure 5 indicates that the average latency rises with an increase in both the transaction rate and the number of transactions. Furthermore, upon analyzing equivalent transaction number groups, it is noteworthy that the delay does not exhibit a significant increase, even as the transaction rate rises. It is evident that the system's throughput and latency could be further enhanced through parameter tuning or the development of optimized Smart Contracts.

### Scenario 2

The second phase aims to evaluate the scalability of healthcare data stored in IPFS. It consists of the data size and the duration of uploading and downloading the data in seconds. For analysis, the data sets used are randomly generated public text files. Figure 6 indicates that the data size spans from 0.003 to 100 MB. Notably, the figure reveals that as the data size expands, both the uploading and downloading times for the data also increase.

### Scenario 3

The third phase involves a comparative analysis between the performance indicators of AguHyper and the experiment data presented in *Kaur, Rani & Kalra (2022)*, *Chelladurai & Pandian (2021)* and *Chelladurai, Pandian & Ramasamy (2021)*. The comparison is conducted based on the settings outlined in Table 3. The primary objective of this phase is to assess the impact of different consensus protocols on system performance by measuring throughput in transactions per second (tps) and average latency in seconds.

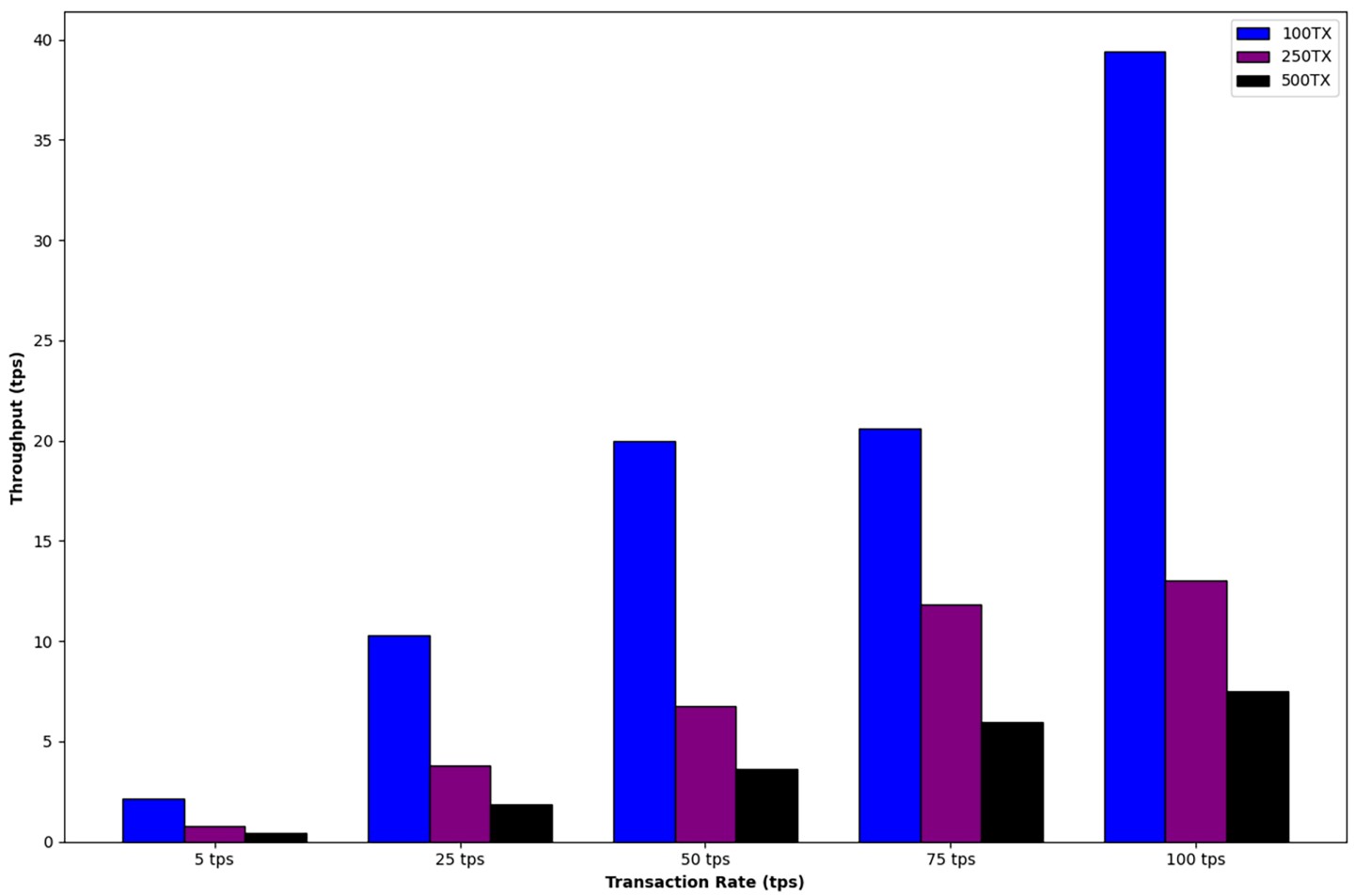

**Figure 4** The influence of altering the number of transactions (Tx) and rate (TPS) on throughput.

According to system configuration and simulation parameters for phase 3, the performance comparison of existing related works (*Kaur, Rani & Kalra, 2022*; *Chelladurai & Pandian, 2021*; *Chelladurai, Pandian & Ramasamy, 2021*) and the proposed work is demonstrated in Tables 4 and 5 based on throughput and average transaction latency. Table 4 shows that the proposed system performs better than the studies by *Chelladurai & Pandian (2021)* and *Chelladurai, Pandian & Ramasamy (2021)* for all transaction groups. It also outperforms the study by *Kaur, Rani & Kalra (2022)*; in the 100, 200, and 500 transaction groups. As a result of Table 5, it is observed that the average transactional latency of the proposed system is marginally higher than the existing works.

The systems under comparison utilize the SOLO consensus mechanism, whereas the proposed system employs the Raft consensus mechanism. Phase 3 experiments were conducted under identical conditions to the existing systems. Therefore, it can be inferred that the utilization of Raft instead of SOLO contributes to an increase in both system throughput and latency.
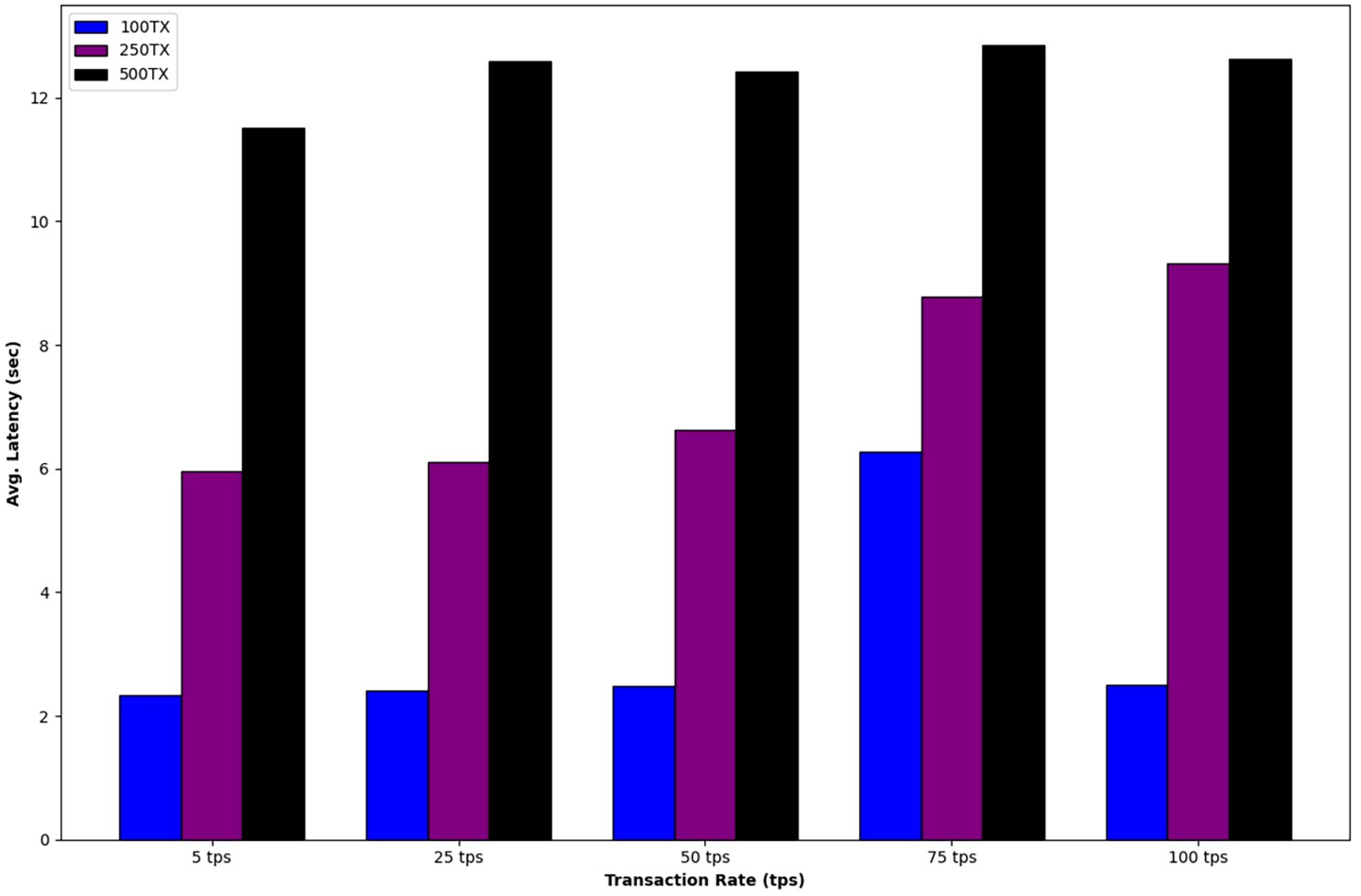

**Figure 5** **The influence of altering the number of transactions (Tx) and rate (TPS) on latency.**

### Scenario 4

In the fourth phase, our objective is to evaluate the correlation between various performance indicators of AguHyper and the experiment data presented in *Sonkamble et al. (2023)*. This assessment is carried out in accordance with the settings specified in Table 6. The fourth phase aims to evaluate the impact of different consensus protocols and state databases on overall system performance by measuring uploading and downloading times.

As per the system configuration and simulation parameters for Phase 4, the performance comparison between the proposed work and existing related work (*Sonkamble et al., 2023*) is depicted in Figs. 7 and 8, focusing on uploading and downloading time. The uploading time encompasses the duration required for uploading data of a fixed size, including its encryption time. On the other hand, downloading time encompasses the total time for downloading the fixed data and the time required for its decryption. Figures 7 and 8 illustrate that the data size ranges from 0.003 to 100 MB.
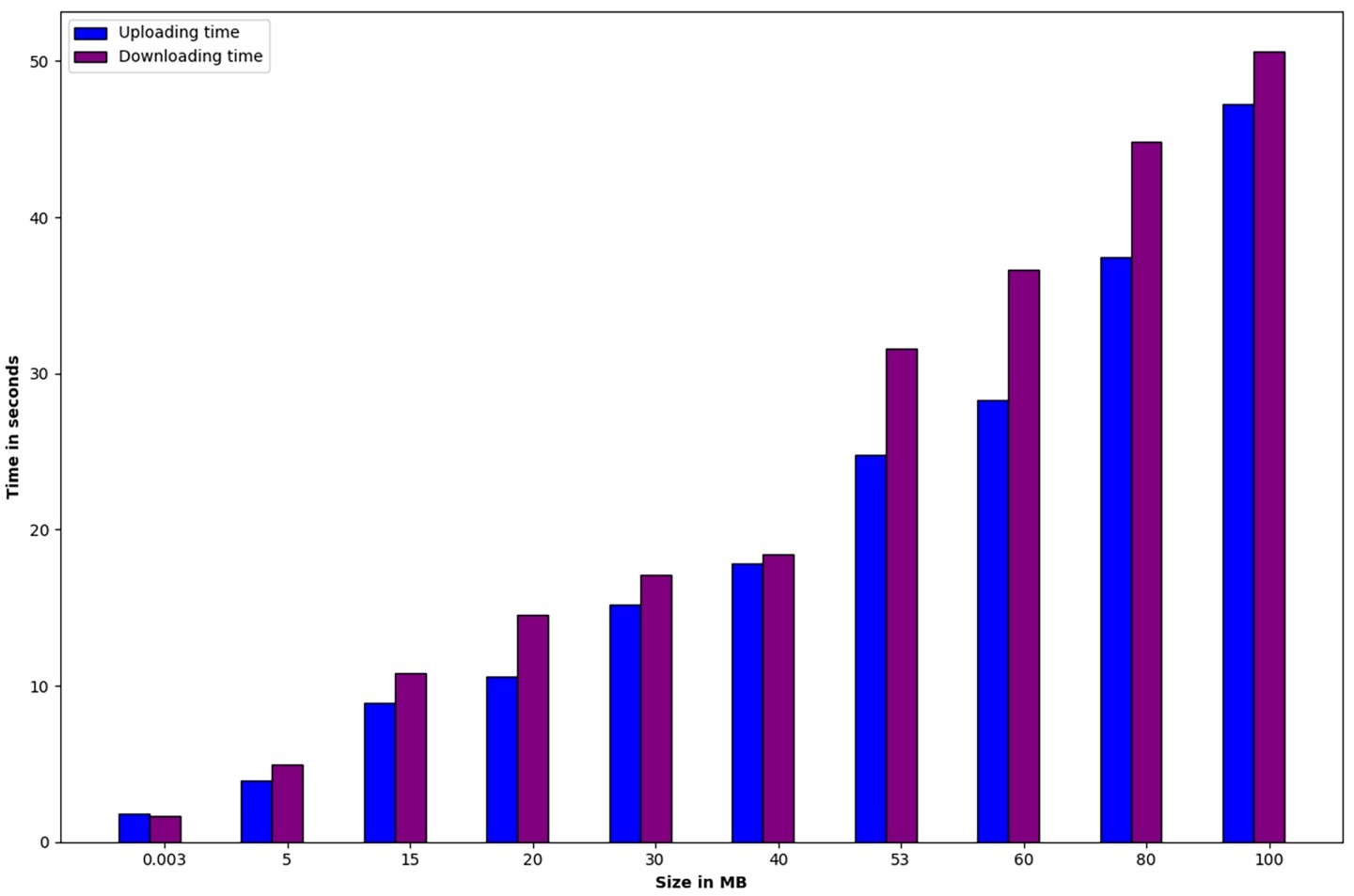

**Figure 6** The process of uploading and downloading EHR data using IPFS.

Specifically, as the data size increases, both uploading and downloading times increase. However, it is observed that the rate of increase in downloading time is higher than that of the uploading time with the increase in block size. The system under comparison utilizes the SOLO consensus mechanism and LevelDB, whereas the proposed system employs the Raft consensus mechanism and CouchDB. Phase 4 experiments were conducted under identical conditions to the existing system. Therefore, it can be inferred that the utilization of Raft and CouchDB instead of SOLO and LevelDB contributes to a decrease in both uploading and downloading time.

### Scenario 5

In the final phase, we conduct a feature-based comparison between AguHyper and existing works based on the ten different questions: Do the studies: i) use access control mechanisms?, ii) explain system permissions?, iii) use data verification mechanisms?, iv) solve security and privacy issues?, v) explain user roles in detail?, vi) use data sharing

**Table 3 System configuration and simulation parameters for phase 3.**

| Phase 3: | Configuration |
| --- | --- |
| Processor | Intel core-i9-9900K-16 CPU |
| Memory | 32 GB |
| OS | Ubuntu 18.04 |
| Hyperledger fabric | v1.4 |
| Transactions | 100, 200, 300, 400 and 500 |
| State DB | CouchDB |
| Orderer and size | AguHyper: Raft and 2 Org-1peer each<br>**Compared works (*Kaur, Rani & Kalra, 2022*; *Chelladurai & Pandian, 2021*;** *Chelladurai, Pandian & Ramasamy, 2021*): SOLO and 2 Org-1peer each |

**Table 4 Phase 3, a performance comparison between the proposed work and existing related works (*Kaur, Rani & Kalra, 2022*; *Chelladurai & Pandian, 2021*; *Chelladurai, Pandian & Ramasamy, 2021*) are conducted based on throughput.**

| Transaction groups (Throughput) | AguHyper | *Kaur, Rani & Kalra (2022)* | *Chelladurai & Pandian (2021)* | *Chelladurai, Pandian & Ramasamy (2021)* |
| --- | --- | --- | --- | --- |
| 100 | 37.6217 | 36.1 | 4.2 | 5.82 |
| 200 | 39.67 | 39.5 | 10 | 10.54 |
| 300 | 34.8397 | 40.9 | 12 | 14.57 |
| 400 | 37.0006 | 40.1 | 16 | 17.89 |
| 500 | 38.1500 | 37 | 20.73 | 21.73 |

**Table 5 Phase 3, a performance comparison between the proposed work and existing related works (*Kaur, Rani & Kalra, 2022*; *Chelladurai & Pandian, 2021*; *Chelladurai, Pandian & Ramasamy, 2021*) are conducted based on average latency.**

| Transaction groups (Average latency) | AguHyper | *Kaur, Rani & Kalra (2022)* | *Chelladurai & Pandian (2021)* | *Chelladurai, Pandian & Ramasamy (2021)* |
| --- | --- | --- | --- | --- |
| 100 | 2.625 | 1.74 | 2.1 | 2.12 |
| 200 | 4.9 | 3.14 | 2.8 | 2.74 |
| 300 | 6.84 | 4.57 | 3.4 | 3.46 |
| 400 | 9.04 | 5.32 | 4.2 | 4.28 |
| 500 | 11.23 | 5.9 | 4.85 | 4.81 |

mechanism?, vii) solve scability issue?, viii) provide the availability?, ix) show performance analysis based on BC?, and x) provide the appropriate basis for disease prediction?

A thorough comparison of features between the proposed work and existing related works is provided in "Related Work", and a summary is presented in Table 1. In contrast to prior research, our proposed solution primarily enables the utilization of EHRs and ensures the secure sharing of these data. We assure information confidentiality, integrity, and optimal data transmission rates across all aspects.

**Table 6 System configuration and simulation parameters for phase 4.**

| Phase 4: | Configuration |
| --- | --- |
| Processor | Intel core-i9-9900K-16 CPU |
| Memory | 32 GB |
| OS | Ubuntu 18.04 |
| Hyperledger fabric | v1.4 |
| Data size | 0.003, 5, 15, 20, 30, 40, 53, 60, 80, 100 MB. |
| State DB | **AguHyper:** CouchDB<br>**Compared work (*Sonkamble et al., 2023*):** LevelDB |
| Orderer and size | **AguHyper:** Raft and 1 Org-3peer<br>**Compared work (*Sonkamble et al., 2023*):** SOLO and 1 Org-3peer |

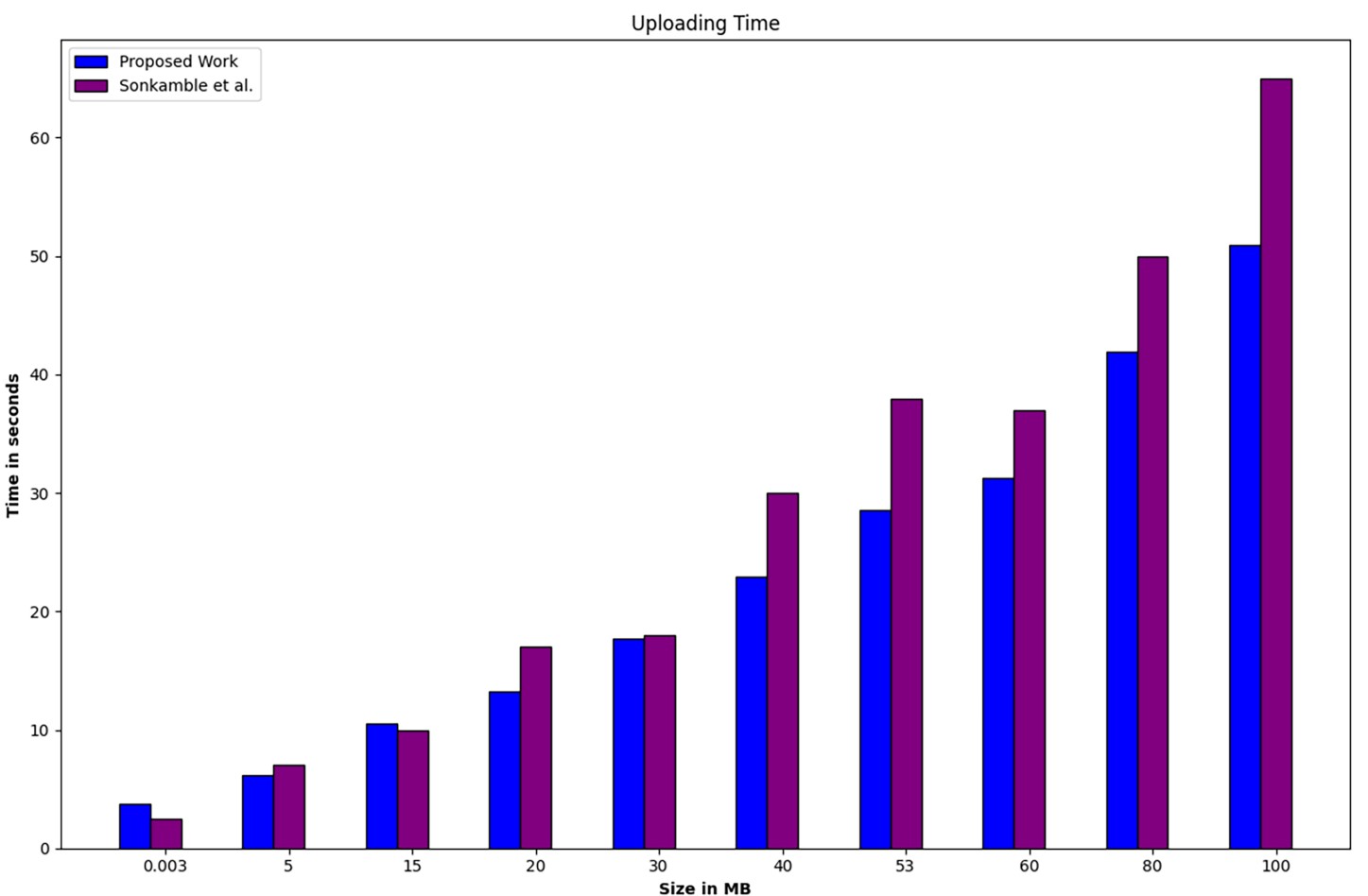

**Figure 7 Phase 4, a performance comparison between the proposed work and existing related work (*Sonkamble et al., 2023*) is conducted based on uploading time.**

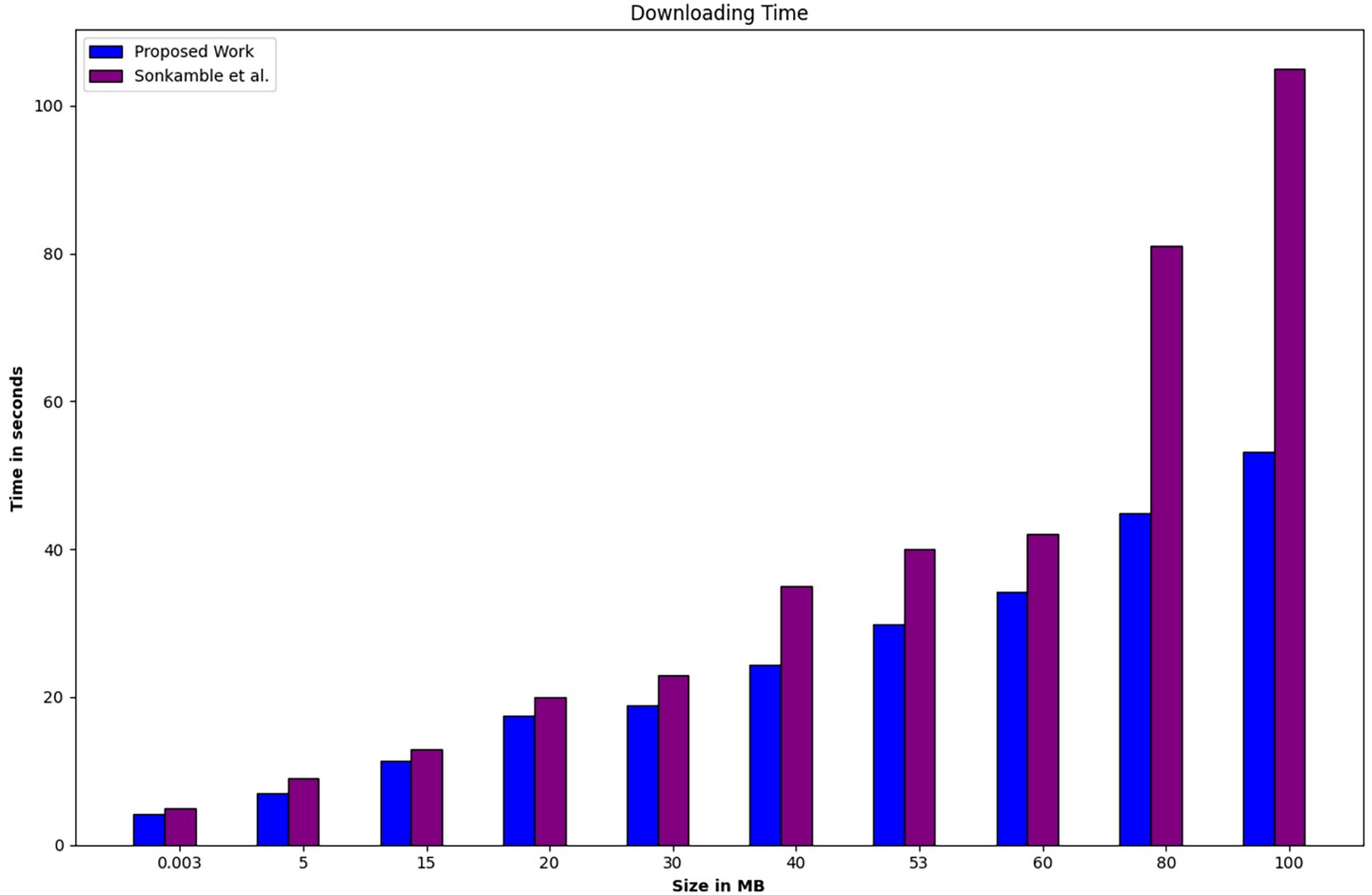

**Figure 8** Phase 4, a performance comparison between the proposed work and existing related work (*Sonkamble et al., 2023*) is conducted based on downloading time.

## CONCLUSIONS

In this article, we propose a permissioned framework built on the Hyperledger blockchain to facilitate the secure sharing and privacy preservation of EHRs. The suggested framework integrates IPFS as a distributed storage solution for EHRs, ensuring that encrypted patient records are securely stored to thwart unauthorized access and malicious attacks. Hash values linked to these records are then embedded in the blockchain distributed ledger. Through the implementation of Smart Contracts (SCs), patients are endowed with comprehensive control over their records, enabling them to grant or revoke permissions to requesters *via* the SCs. All transactions are meticulously recorded on the immutable and decentralized blockchain ledger. The study conducts in-depth analyses of the system architecture, AguHyper implementation configurations, and meticulous performance evaluations using diverse datasets. The experimental setup incorporates CouchDB and the Raft consensus mechanism, with the system's performance scrutinized in terms of throughput and latency. This comparison against existing studies contributes to a

thorough and comprehensive assessment. Importantly, this investigation introduces a distinctive perspective to the existing literature in the field.

The findings of the analysis indicate that the suggested solution is pragmatic and adeptly fulfills a variety of security requisites. It manifests noteworthy promise in safeguarding the security, privacy, confidentiality, integrity, and scalability of health data. Future improvements may focus on enhancing the framework's functionality to provide quicker responses to queries, thereby reducing response time, latency, and overall costs. Furthermore, there is an objective to expand the framework's coverage to encompass additional data sharing scenarios. Potential future works could explore advanced encryption techniques to further fortify data security, as well as the integration of artificial intelligence algorithms for predictive analysis and anomaly detection within the EHR system. These endeavors will contribute to the continued evolution and refinement of AguHyper, fostering its adoption and relevance in the dynamic landscape of healthcare data management.

### Funding
The authors received no funding for this work.

### Competing Interests
Burcu Bakir-Gungor is an Academic Editor for PeerJ

### Author Contributions
- Beyhan Adanur Dedeturk conceived and designed the experiments, performed the experiments, analyzed the data, performed the computation work, prepared figures and/or tables, and approved the final draft.
- Burcu Bakir-Gungor conceived and designed the experiments, prepared figures and/or tables, authored or reviewed drafts of the article, and approved the final draft.

### Data Availability
The code file is available in the Supplemental File.

The data is available at Zenodo: adanur dedeturk,. beyhan. (2023). Sized Files [Data set]. Zenodo. https://doi.org/10.5281/zenodo.10251250.

### Supplemental Information
Supplemental information for this article can be found online at http://dx.doi.org/10.7717/peerj-cs.2060#supplemental-information.

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
