# Peer review of "Aguhyper: a hyperledger-based electronic health record management framework"

_PeerJ Computer Science, doi:10.7717/peerj-cs.2060_

## Round 0.1 · original submission · Major Revisions

Dear Authors,

We have received the reviews for your manuscript. The reviewers have provided valuable feedback, and it is evident that major revisions are needed to address the concerns raised. Below, I have summarized the critical points raised by each reviewer:

The introduction lacks highlighting the research significance and needs to be more concise.

Language improvements are necessary for better clarity and understanding by an international audience.

Figures and tables should be appropriately placed and cited.

Feasibility of the proposed scheme is questioned, particularly regarding security and functional analysis.

Management of public-private key pairs needs clarification.

Various sections need improvement, including the Abstract, Introduction, Related Work, and Experimental Setup.

The structure of the paper needs enhancement, along with the inclusion of recent related schemes and citations.

Technical details, algorithm formatting, dataset source, results generation, training process, novelty identification, equation key terms, software for diagrams, paper organization, and English language need improvement.

The importance of secure electronic storage and sharing of healthcare records is acknowledged.

Specific references are requested for certain statements.

Clarifications are needed regarding the methodology, blockchain definition, and references for literature examined.

Based on these critical points, I recommend a major revision of the manuscript. Please carefully address each concern raised by the reviewers in your revised submission. Ensure clarity, conciseness, and thoroughness in your revisions. Additionally, pay close attention to language quality, technical details, and the organization of your paper.

Feel free to reach out if you have any questions or need further clarification on the reviewers' comments. We look forward to receiving your revised manuscript.

**Language Note:** The Academic Editor has identified that the English language must be improved. PeerJ can provide language editing services - please contact us at [email protected] for pricing (be sure to provide your manuscript number and title). Alternatively, you should make your own arrangements to improve the language quality and provide details in your response letter. – PeerJ Staff

Reviewer 1 ·

Basic reporting

The author proposed a secure AguHyper scheme to store and share the EHRs, based on the IPFS and blockchain technology. However, serveral problems need to be addressed.
1) In the introduction section, the research significance of the manuscript is not highlighted; In addition, the content should be concise.

2)The reference is insufficient,some key references are missing, please try to add some recent findings in this area, such as:
[1] A Novel Patient-Centric Architectural Framework for Blockchain-Enabled Healthcare Applications[J].IEEE Transactions on Industrial Informatics, 2020.DOI:10.1109/TII.2020.3037889.
[2]B-DSPA: A Blockchain-Based Dynamically Scalable Privacy-Preserving Authentication Scheme in Vehicular Ad Hoc Networks, in IEEE Internet of Things Journal, 2024. doi: 10.1109/JIOT.2023.3289057.

3) The English language should be improved to ensure that an international audience can clearly understand your text. Check and correct grammar problems in the manuscript, such as lines 39-41, 114-116.

4)To make it easier for readers,it is recommended that the figure and table be placed in the position cited in the manuscript.

Experimental design

1) The feasibility of the scheme is difficult to determine, and the scheme needs to be improved, such as in the security and functional analysis section, the description at lines 503-509, why the data header informtion is necessary? how to addresses privacy, scalability and accessibility challenges?

2)According the detail at lines 404-406, each user is equipped with a public-private key pair. How to manage the keys?

Validity of the findings

1) The innovation is not obvious, and a lot of studies have been done , as description at lines 815 ( Jayabalan, J., & Jeyanthi, N. (2022)), 822 (Kannan S, Smith M. 2016).

Cite this review as

Reviewer 2 ·

Basic reporting

Please check additional comments.

Experimental design

Please check additional comments.

Validity of the findings

Please check additional comments.

Additional comments

1. Please improve the Abstract.
2. The Introduction section is very poor. In a research article, the introduction section must be very strong with the motivations of this paper, which is missing in this paper. Moreover, the disadvantages of the existing schemes must be discussed to motivate this new work.
3. The given point-wise contributions are not specific. The last paragraph of the Introduction should be the structure of the paper.
4. The Related Work section is poor. The authors must include some more recent schemes. Also, the following papers must be cited to improve this section, as well as the Reference section:
a) Network anomaly detection using deep learning techniques
b) Problem-based cybersecurity lab with knowledge graph as guidance
c) Blockchain-based smart contract model for securing healthcare transactions by using consumer electronics and mobile edge computing
d) Local binary pattern-based reversible data hiding
e) Taxonomy of DNA-based security models
f) Review on offloading of vehicle edge computing
g) Blockchain-based cloud storage system with enhanced optimization and integrity preservation
5. In section 2, a table can be given to summarize the entire section.
6. The BACKGROUND and PRELIMINARIES section can be ignored.
7. How has the performance of the prediction increased?
8. Different layers are given. However, there are no links among them.
9. Algorithm 3 is wrong. Also, algorithms are not properly formatted.
10. Which entities are involved for data management and how?
11. Which consensus algorithm is used in this paper and why?
12. The “Experimental Setup” subsection is not convincing. Add section numbering.
13. What is the source of the dataset? Whether it is authentic or not? Mention clearly.
14. How the results of Figure 5 are generated?
15. Technical details about results are missing.
16. How the training is done?
17. What is the novelty of this work? It is hard to identify from the current version of this paper.
18. Key terms of the equations must be defined.
19. Use a well-known software to draw the diagrams of the results section.
20. The organization of the paper must be improved. The paper must be formatted properly.
21. Improve the English language.
22. The Reference section must be improved significantly.
23. Please give two paragraphs in the last section. One for concluding the entire chapter, and the second one for discussing future works. Also, complete it within 400 words.

Cite this review as

Reviewer 3 ·

Basic reporting

This article highlights the growing importance of secure electronic storage and sharing of healthcare records. Introducing AguHyper, a secure solution built on a permissioned blockchain framework using Hyperledger Fabric and IPFS, the system ensures security, privacy, scalability, and data integrity.

Experimental design

Major comments:

Please, add a reference for: “A considerable obstacle confronting the contemporary healthcare industry lies in the insufficient interoperability of EHRs”;

Please, add a reference for: “The vulnerability to internal attacks, wherein individuals with legitimate credentials gain entry to data, poses a substantial risk to health records stored on cloud servers, surpassing the threats posed by external attacks”;

Please, justify the following sentence: “Despite the commendable features of the existing healthcare industry, it falls short in providing a universally unified and efficient approach for storing, sharing, and analyzing health data”;

I suggest to update some references in the introduction;

“Our proposed methodology called AguHyper integrates Hyperledger Fabric (Androulaki et al. 2018) the IPFS with the goal of exceeding the established efficiency benchmarks and addressing gaps identified in prior studies within this domain” - maybe there is a missing “and”;

Why do the authors define blockchain (BC) only starting from related works?

Please, add references for: “Notably, the blockchain-based platforms in this period cover both genomic data sharing and EHR sharing.”;

I think it is necessary to cite the reference articles for the literature examined from line 181 to 193.



Minor comments:

Pay attention to some repeated acronyms;

Pay attention to some punctuation marks in inappropriate positions.

Validity of the findings

no comment

Cite this review as

---

## Round 0.2 · Minor Revisions

After careful consideration of your comments and suggestions, we have decided to proceed with a major revision of the manuscript. We acknowledge the need for improvements in logic rigor and the inclusion of relevant literature to enrich the discussion.

Here is a summary of the key points raised in your review:

Rigorous Logic: The reviewers highlighted the necessity for a more rigorous logic throughout the manuscript, particularly in substantiating claims made in the "Security Analysis" section. For instance, while the last contribution mentions improved security against DoS attacks, single points of failure, and enhanced data integrity, there is a lack of related proof in the analysis.

Literature Inclusion: The reviewers recommended incorporating additional literature to strengthen the manuscript's context and relevance.

Please address the comments raised by reviewers.

Reviewer 1 has suggested that you cite specific references. You are welcome to add it/them if you believe they are relevant. However, you are not required to include these citations, and if you do not include them, this will not influence my decision.

Reviewer 1 ·

Basic reporting

The author has revised the manuscript according to the reviews.
1) The logic should to be more rigorous. such as the last contribution is detailed "it can provied better security against DoS attacks, single points of failure, and improving data integrity". But in the section security analysis, there is not related proof.
2) It is suggested to add the literature, such as:

a) Hyperledger fabric access control system for internet of things layer in blockchain-based applications,
b) A novel lightweight decentralized attribute-based signature scheme for social co-governance,
c) Blockchain-Enabled and Data-Driven Smart Healthcare Solution for Secure and Privacy-Preserving Data Access

Experimental design

no comment

Validity of the findings

no comment

Cite this review as

Reviewer 2 ·

Basic reporting

All the previous comments are addressed.

Experimental design

NA

Validity of the findings

NA

Additional comments

NA

Cite this review as

Reviewer 3 ·

Basic reporting

The authors have addressed all the reviewers' comments, and the manuscript appears to have significantly improved at this point.

Experimental design

The experimental design has been improved and now the paper is read for the publication.

Validity of the findings

At this point, the reported results could improve the existing literature.

Cite this review as

---

## Round 0.3 · accepted · Accept

Dear Authors,

I am pleased to inform you that your paper has been accepted for publication in our journal.